# Use of Iontophoresis with Corticosteroid in Carpal Tunnel Syndrome: Systematic Review and Meta-Analysis

**DOI:** 10.3390/ijerph20054287

**Published:** 2023-02-28

**Authors:** Francisco Javier Martin-Vega, Maria Jesus Vinolo-Gil, Gloria Gonzalez-Medina, Manuel Rodríguez-Huguet, Inés Carmona-Barrientos, Cristina García-Muñoz

**Affiliations:** 1Department of Nursing and Physiotherapy, University of Cadiz, 11009 Cadiz, Spain; 2Rehabilitation Clinical Management Unit, Interlevels-Intercenters Hospital Puerta del Mar, Hospital Puerto Real, Cadiz Bay-La Janda Health District, 11006 Cadiz, Spain; 3Research Unit, Department Biomedical Research and Innovation Institute of Cadiz (INiBICA), Puerta del Mar University Hospital, University of Cadiz, 11009 Cadiz, Spain; 4CTS-986 Physical Therapy and Health (FISA), University Institute of Research in Social Sustainable Development (INDESS), 11009 Cadiz, Spain

**Keywords:** iontophoresis, corticosteroids, carpal tunnel syndrome, systematic review, meta-analysis

## Abstract

Background: Carpal tunnel syndrome is a neuropathy that affects the median nerve. The aim of this review is to synthesize the evidence and perform a meta-analysis on the effects of iontophoresis in people with carpal tunnel syndrome. Methods: The search was carried out using PubMed, Web of Science, Scopus, CINHAL Complete, Physiotherapy Evidence Database, and SciELO. The methodological quality was evaluated using PEDro. A standardized or mean difference meta-analysis (Hedge’s g) using a random-effects model was calculated. Results: Seven randomized clinical trials using iontophoresis for electrophysiological, pain, and functional outcomes were included. The mean of PEDro was 7/10. No statistical differences were obtained for the median sensory nerve conduction velocity (SMD = −0.89; *p* = 0.27) or latency (SMD = −0.04; *p* = 0.81), motor nerve conduction velocity (SMD = −0.04; *p* = 0.88) or latency (SMD = −0.01; *p* = 0.78), pain intensity (MD = 0.34; *p* = 0.59), handgrip strength (MD = −0.97; *p* = 0.09), or pinch strength (SMD = −2.05; *p* = 0.06). Iontophoresis only seemed to be superior in sensory amplitude (SMD = 0.53; *p* = 0.01). Conclusions: Iontophoresis did not obtain an enhanced improvement compared to other interventions, but no clear recommendations could be made due to the limited number of included studies and the heterogeneity found in the assessment and intervention protocols. Further research is needed to draw sound conclusions.

## 1. Introduction

Carpal tunnel syndrome (CTS) is one of the most common neuropathies in the upper limb and affects the median nerve in the wrist area [1,2]. The prevalence of CTS has been estimated to be around 5% in the general population [3].

The incidence of CTS has been estimated as 276 cases per 100,000 inhabitants [1,4]. Symptoms may include a sensation of tingling, numbness, and paresthesia, as well as pain and weakness in the hand area, which affects the handgrip and pinch strength [5,6]. In addition, electrophysiological disorders in conduction velocity, amplitude, and latency could be found in CTS [7,8]. Considering all this, CTS has repercussions in the psychophysical, social, and labor spheres [9]. Rehabilitation and surgery are the principal therapeutic approaches in people diagnosed with CTS. Within the scientific literature, the other therapeutic options that have been systematically reviewed in CTS include: shock waves (based on high energy waves) [10], phonophoresis (based on ultrasounds) [11], or injections (use of steroids, cortisone) [12]. One of the therapeutic options for this condition that needs further study is iontophoresis. Iontophoresis is a therapeutic approach involving transdermic administration of a drug (corticosteroids), using a continuous or galvanic electric current to promote ion migration through the skin [13].

Previous research focused on the use of corticosteroid in neuropathies [14], as well as the use of different modalities of electrotherapy in people diagnosed with CTS [15]. Other studies compared the use of iontophoresis with surgery-based interventions [12]. However, to our knowledge, we did not find any systematic reviews that specifically assessed the use of corticosteroids combined with iontophoresis in people diagnosed with CTS. Therefore, we conducted a systematic review with meta-analysis to synthesize the evidence and analyze the effects of iontophoresis compared to other interventions in people with CTS.

## 2. Materials and Methods

### 2.1. Study Design

To conduct this systematic review with meta-analysis we followed the Preferred Reporting Items for Systematic Reviews and Meta-Analyses (PRISMA) 2020 statement. and the PRISMA for abstracts [16]. A prospective registry of the study was performed in the Open Science Framework registry with the DOI: https://doi.org/10.17605/OSF.IO/2SUH7.

### 2.2. Search Strategy

To build the search strategy we combined terms related to the assessed intervention (“Iontophoresis” (MeSH) and Treatments), and to the disease condition (“Carpal Tunnel Syndrome” (MeSH) and “Median Neuropathy” (MeSH)). The mentioned terms were combined using the Boolean operator AND, as Table 1 shows. The search was conducted by two independent reviewers in the PubMed database, Web of Science (WoS), Expertly curated abstract and citation database (Scopus), Cumulative Index to Nursing and Allied Health Literature (CINHAL), Physiotherapy Evidence Database (PEDro), and Scientific Electronic Library Online (SciELO). No filters were used. The search was performed from the databases’ inception.

### 2.3. Eligibility Criteria

To develop the inclusion and exclusion criteria, we used the PICOS model (Population, Intervention, Outcomes, Study design) [17]. The inclusion criteria were as follows: P: participants diagnosed with carpal tunnel syndrome; I: iontophoresis with corticosteroid; C: any type of comparator; O: pain, functionality, and electrophysiological variables (e.g., velocity, amplitude) and S: Randomized clinical trials. The exclusion criteria were studies not conducted in humans.

### 2.4. Study Selection and Data Extraction

The search, screening, and selection of the potential studies were conducted by two independent reviewers. After a selection based on title and abstract, the full text was retrieved to assess if the records met the selection criteria. Before that, duplicate records were manually excluded using the Mendeley desktop software (v. 1.19.8). When there was a disagreement, a third reviewer was consulted for consensus.

To perform descriptive and quantitative analysis, we extracted from each study the information relating to the author, year and study design, sample characteristics, outcomes, experimental and control interventions, intervention parameters, and main findings of the studies.

### 2.5. Methodological Quality Assessment

We used the PEDro assessment scale to examine the methodological quality of the included randomized clinical trials [18]. This scale consists of 11 items, but only 10 are computed for the final score (the first item is not considered). Level I of evidence is achieved with a score equal or higher to 6 points (6–8: good and 9–10: excellent), while a score equal or lower than 5 (4: poor and 4–5: regular) indicates a Level II of evidence [19].

### 2.6. Meta-Analysis

To run the meta-analyses, the R studio software (v. 4.2.1) and the packages meta (v. 5.1-1), metafor (v. 3.0-2), and dmetar (v. 0.0.9000) were used. Assuming heterogeneity between the included studies, the available data were pooled using a random-effects model and an inverse variance method. We calculated the standard mean differences (SMD) when the outcome was assessed differently between studies and when the same measure was employed, we performed a mean difference meta-analysis (MD). We calculated the estimated effect size Hedge’s g method and its corresponding 95% confidence intervals (95% CI). Hedges’s g score below 0.2 indicated a small effect size, while those above 0.5 were medium or large when >0.8. When a study presented more than two arms of study, independent comparisons were included in the analysis.

We assessed heterogeneity between studies through the score of I^2^ showing substantial heterogeneity when I^2^ > 50% and the tau-square test. This information was added to the forest plots, as well as the prediction interval to predict a future observation based on the given data. A sensitivity analysis was carried out for each meta-analysis to detect potential outliers. A sensitivity analysis was only conducted in those meta-analyses that included at least three studies. Automatic detection using the dmetar package, influence analysis, and leave-one-out analysis sort by effect size were used for the sensitivity analysis. When a study was identified as an outlier, we removed it from the meta-analysis. In addition, publication bias was examined through the Egger’s bias test and funnel plots were used for the graphical representation. Statistically, Egger’s tests could only be conducted when there were at least three studies included in the meta-analysis.

### 2.7. Certainty of Evidence

Two independent reviewers rated the certainty of evidence obtained in the meta-analysis findings using the GRADE approach assessment tool [20]. The level of evidence was evaluated based on the risk of bias, inconsistency of results, indirectness of evidence, imprecision, and publication bias. Finally, the level of evidence could be scored as ‘high’, ‘moderate’, ‘low’, or ‘very low’. Considering that all the included studies were randomized controlled trials, the level of evidence started with a ‘high’ score that would be downgraded depending on the results of the mentioned factors.

## 3. Results

### 3.1. Studies Selection and Studies Characteristics

A total of 296 records were obtained from the search in the different databases, with only 7 randomized clinical trials meeting the selection criteria [21,22,23,24,25,26,27] (Figure 1).

The main characteristics of the included studies are shown in Table 2. The total sample size was 262 participants with a mean ranging from 30 years in the study of Elrakiz et al. [21] and 54 years old in the study of Amirjani et al. 2009 [26]. Among the studies, 90% had samples where women predominated against men [21,23,26,27]. In the four studies that reported bilateral cases [22,25,26,27], 67% of the participants met this condition. All randomized clinical trials reported a mild or moderate level of severity, but two studies did not report this data [21,24].

The electrophysiological outcomes measured were median nerve sensory and motor conduction velocity, amplitude, and distal latency [21,22,23,25,26,27]. Median nerve motor conduction velocity, amplitude, and distal latency were also evaluated [28]. Handgrip and pitch strength were assessed using dynamometry [22,23,24,25]. The visual analogue self-assessment scale (vas) was used to measure perceived pain intensity [21,22,23,25]. The Boston Carpal Tunnel Questionnaire (BCTQ) [24,26,27] was used to examine the functional and sensory severity of CTS [29]. The included studies evaluated somatosensorial disorders [23,26] using the Semmes–Weinstein Test [30]. The Phalen and Tinnel test [23] specifically evaluated the CTS [31]. The Nine-Hole Peg Test [24] assessed functional and hand dexterity [32]. Finally, the Boston Questionnaire (BQ) and the Health Assessment Questionnaire (HAQ) [23] evaluated the mobility during wrist flexo–extension movements, muscle strength, and disability during activities of daily living.

The drugs combined with iontophoresis were dexamethasone sodium phosphate [22,25,26,27], dexamethasone sodium diphosphate [23], dexamethasone with lidocaine [21], and betamethasone [24].

The iontophoresis parameters ranged from an intensity of 1 mA per minute [22,26,27] to 4 mA per minute [24,27]. Administration time ranged from 10 min per session [24,25,27] to 20 min [21,22,23]. The total number of sessions ranged from 6 sessions [26] to 18 [21].

### 3.2. Meta-Analysis

#### 3.2.1. Electrophysiological Outcomes

Median sensory nerve conduction velocity

We performed a meta-analysis on a total of three studies, two with three arms of study [21,23,27], and found a lack of statistical significance SMD = −0.89 (*p* = 0.27) and 95% CI (−2.82 to 1.05) and a high heterogeneity between the studies (I^2^ = 87%; Tau^2^ = 2.09) (Figure 2). After the sensitivity analysis, one [21] study was identified as an outlier, changing the findings to SMD = −0.24 (*p* = 0.24), 95% CI (−0.77 to 0.28) with an I^2^ = 0% and Tau^2^ = 0 (Appendix B). No publication bias was detected in the funnel plot or Egger test = 0.18. The GRADE tool assessment classified this finding as ‘very low’ (Appendix C).

Median Sensory nerve amplitude

The studies of Aygül et al. [27] and Duymaz et al. [23] with all their intervention arms were pooled. The iontophoresis groups showed a significant difference for the improvement in median sensory amplitude after intervention than the comparator group with no significant differences between groups (SMD = 0.51 (*p* = 0.02); 95% CI (0.19 to 0.84); I^2^ = 0% and Tau^2^ = 0) (Figure 3). The GRADE tool assessment classified this finding as ‘very low’.

Median sensory nerve distal latency

The studies of Aygül et al. [27] and Duymaz et al. [23] with all their intervention arms were pooled. The iontophoresis groups showed a significant difference for the improvement in median sensory amplitude after intervention than the comparator group with no significant differences between groups (SMD = −0.04; *p* = 0.81); 95% CI (−0.52 to 0.44); I^2^ = 0% and Tau^2^ = 0) (Figure 4). The GRADE tool assessment classified this finding as ‘very low’.

Median Motor nerve conduction velocity

We performed meta-analysis on two studies [23,27], both with three arms of interventions, and found no differences in median motor nerve conduction velocity between the intervention groups (SMD = −0.04 (*p* = 0.88); 95% CI (−0.74 to 0.66); I^2^ = 26% and Tau^2^ = 0.056) (Figure 5). The GRADE tool assessment classified this finding as ‘very low’.

Median Motor nerve latency

We performed meta-analysis on two studies [23,27] and found no differences in median motor nerve latency between iontophoresis and the other interventions (SMD = −0.01 (*p* = 0.78); 95% CI (−0.13 to 0.10); I^2^ = 0% and Tau^2^ = 0) (Figure 6). The GRADE tool assessment classified this finding as ‘very low’.

#### 3.2.2. Pain Intensity

Visual analog scale

We performed meta-analysis on three studies [21,23,25], one of them multi-arm [23]. As is shown in the forest plot (Figure 7) no differences between experimental and control interventions were found (*p* = 0.59) with an effect size of SMD = 0.34 and 95% CI (−1.27 to 1.95). The heterogeneity between studies was high (I^2^ = 0% and Tau^2^ = 0). After the sensitivity analysis, no outlier was detected. The GRADE tool assessment classified this finding as ‘very low’.

#### 3.2.3. Functional Outcomes

Handgrip strength

Only two randomized controlled trials were included in the meta-analysis, showing no significant differences between phonophoresis compared to iontophoresis (MD = −0.97 (*p* = 0.09); 95% CI (−2.63 to 0.70); I^2^ = 0% and Tau^2^ = 0) (Figure 8). The GRADE tool assessment classified this finding as ‘very low’.

Pinch strength

As the forest plot in Figure 9 shows, iontophoresis showed a trend to be less effective than phonophoresis in improving pinch strength, but the pooled results did not indicate differences between interventions (MD = −2.05 (*p* = 0.06); 95% CI (−4.39 to 0.29); I^2^ = 0% and Tau^2^ = 0). The GRADE tool assessment classified this finding as ‘very low’.

## 4. Discussion

This systematic review with meta-analysis aimed to synthesize the effects of iontophoresis combined with corticosteroids compared to other interventions in people diagnosed with CTS. No differences were found between iontophoresis-based interventions compared to the comparator groups for pain intensity and electrophysiological outcomes (except for the median sensory nerve amplitude) and handgrip and pinch strength.

Regarding the severity of CTS, most of our analyzed studies focused on mild to moderate cases, as in the review of Martín-Vega FJ et al. [11], for phonophoresis. This fact could be related to the fact that most severe cases used to be treated with invasive approaches or surgery [33].

In the included studies, no consensus was reached for the effects of iontophoresis against other therapeutic approaches. Some studies reported higher effects of phonophoresis than iontophoresis for pain reduction, higher handgrip or pinch strength, sensory conduction velocity, and median motor nerve parameters [22,25]. However, Gurkay et al. [24] did not report differences between the mentioned interventions. When iontophoresis was compared to local injection of corticosteroids in the study by Aygül et al. [27], the control intervention showed better results for median sensory nerve conduction velocity. Dexamethasone and betamethasone were the main drugs used in our included trials, which agrees with the previous review of Martín-Vega et al. [11] that examined the effects of phonophoresis in the same target population. The use of these drugs has been reported to be effective for pain and inflammatory processes [34,35].

When iontophoresis combined with corticosteroids was compared to controls without pharmacological treatments, it was superior to ultrasounds [23] but not to shock waves [21]. Shock wave-based therapy has been found to be effective in people with CTS, but, as in the case of iontophoresis, a large number of randomized controlled trials are needed [10]. Additionally, one study [26] did not record differences when the iontophoresis was compared to placebo. Considering the previous information, further research is needed to know the real effects of iontophoresis combined with corticosteroids in people with CTS. Due to the limited number of previous systematic reviews and meta-analyses that evaluated the effects of iontophoresis plus corticosteroids, it is difficult to establish comparisons of our findings with the one reported by other authors.

Only the median sensory nerve amplitude showed significant effects with iontophoresis compared to local steroid injections, phonophoresis, ultrasounds, and sham iontophoresis. Nonetheless, this finding could change with future studies as the prediction interval indicated. Further randomized clinical trials are needed to know whether iontophoresis is a suitable therapeutic option to improve the amplitude of sensory nerves and to understand the mechanism behind this effect.

Regarding pain, iontophoresis was shown to be effective in other diseases [36,37], but only a short term effect was found [38]. Most of trials in our review showed effective iontophoresis with this therapeutic goal in mild to moderate cases of CTS. However, no therapeutic supremacy was recorded in the meta-analysis against comparator interventions [21,22,23,24,25,27]. This is in line with the results reported by Martín-Vega et al. [11], in which the effects of iontophoresis plus corticosteroids showed similar results to phonophoresis with corticosteroids [22,24,25,27].

Although our meta-analysis did not only focus on iontophoresis versus phonophoresis in the case of handgrip and pinch strength, this pairwise comparison happened. These meta-analyses did not record significant differences between the interventions, but a trend favoring phonophoresis was found. The latter statement is in line with the reviews of Huisstede et al. [15] and Martín-Vega et al. [11] that declared major improvements in interventions based on phonophoresis rather than iontophoresis. Nonetheless, these studies supported that further research was needed to confirm the real effects between these therapeutic approaches. 

The lack of consensus in the treatment parameters of iontophoresis and throughout the assessment process is plausible throughout the included studies. The heterogeneity found in the primary studies could bias our results, which was the primary reason we rated the certainty of evidence of our findings as very low using the GRADE approach. In addition to this, the scarce number of primary studies included in the meta-analysis makes it impossible to provide sound clinical recommendations. Our request to follow a standardized protocol agrees with previous literature [1,11,39]. Regarding the diagnosis or assessment, electrophysiological testing is the most popular in the scientific literature [11], because of its high reliability and feasibility [40]. Despite this, the electrophysiological assessment must be supported with other evaluation tools or procedures [2,41]. Concerning the treatment parameters of iontophoresis, Anderson et al. [42] suggested the use of low intensity and a longer time to achieve a larger duration and effectiveness of the administration of dexamethasone through iontophoresis. 

### Strengths and Limitations

This systematic review has strengths. The methodological quality of the included randomized clinical trials assessed using the PEDro scales was good, but this does not exclude the presence of biases that could influence our results. We did not use language filters, which could limit the inclusion of potential bias; despite this, some of the results of the included studies could have been misinterpreted due to the language barriers of the authors. The performance of meta-analyses is a strength of this review because it is a first step in examining the effects of iontophoresis plus corticosteroid in the current literature. In addition, the use of the GRADE approach is an additional advantage of our study, allowing us to provide a level of certainty for our analysis. Furthermore, the evaluation of our findings through GRADE paves the way for substantiated recommendations to improve the quality of future research. One of our major recommendations extracted from the GRADE evaluation and after synthesis of studies is the real need of a standardized protocol for measurement of the outcomes, like those provided in the Guidelines in electrodiagnostic medicine of the American Association of Electrodiagnostic Medicine [28], and iontophoresis-based parameters. Considering all this, a call for action can be extracted from our results: the need for a standardization in reporting data, and proposed a clear protocol of study would help in obtaining future sound results. This would allow for robust clinical recommendations to be made in the future.

This review also has limitations. As we mentioned before, the heterogeneity found in the primary studies could influence our findings, and should be considered with caution. One perfect example of this is the meta-analysis conducted for the median sensory conduction velocity in which the heterogeneity between studies even after the sensitivity analysis remained substantially high. Some of our meta-analyses could only be performed using post-intervention mean and standard deviation, because, in the electrophysiological outcomes in particular, most authors did not carry out a baseline analysis. This could influence the interpretation of the findings of the primary studies and, therefore, ours. Furthermore, we could not perform the meta-analysis sort by the same type of comparator groups because of the heterogeneity of primary studies. A major limitation of these articles used in the meta-analysis is that none of them used control groups without treatment or placebo control groups, so no strong clinical conclusions could be drawn. Therefore, it cannot be said that the treatments have a positive effect. Another limitation related to our meta-analyses that should be considered is the limited number of primary studies included, so the sample size, as we reported in the GRADE table, could bias our external validity. However, as we previously mentioned, our analysis is a first step in this research topic.

## 5. Conclusions

Therapeutic approaches based on iontophoresis combined with corticosteroids achieved similar results for pain intensity and electrophysiological outcomes, except for median sensory nerve distal latency, when compared to other interventions. The latter finding should be considered with caution due to the scarce number of primary studies that examined this outcome and the variability between the sample size and the assessment between randomized controlled trials. The handgrip and pinch strength showed better improvements in the group that received phonophoresis than those who received iontophoresis, but the pooled results did not show differences between interventions.

As we reported in the GRADE approach, our findings should be interpreted with caution, so no clear clinical recommendation could be provided. However, after the synthesis of the available literature, we claim the need of a standardization of the assessments of participants, as well as in the protocols of the interventions. All these, will allow us to achieve sound findings and to propose clear clinical recommendations.

## Figures and Tables

**Figure 1 ijerph-20-04287-f001:**
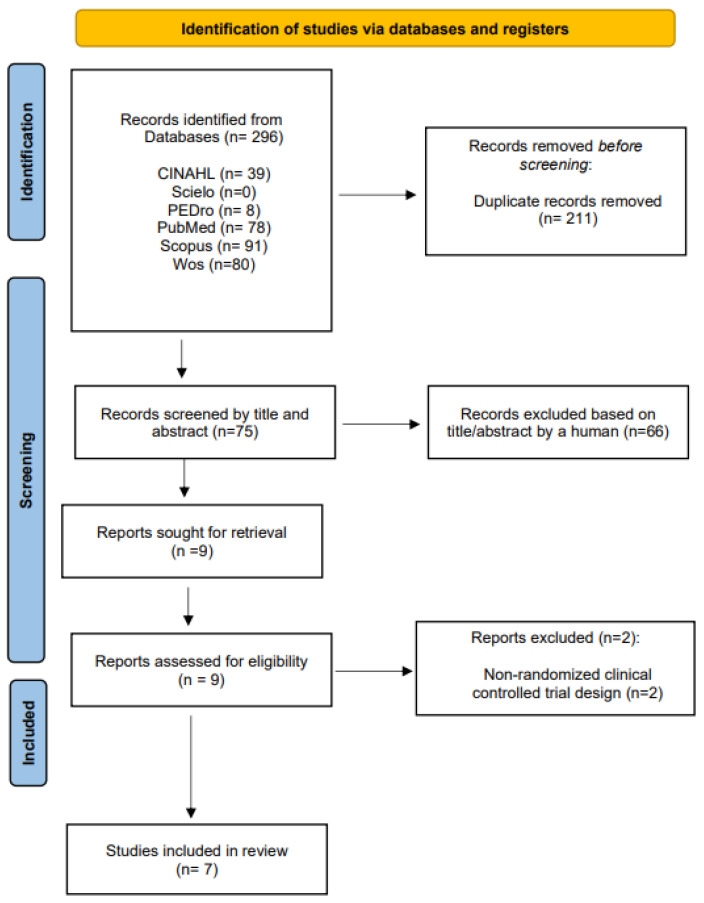
Flow diagram of trial selection based on the PRISMA 2020 guidelines.

**Figure 2 ijerph-20-04287-f002:**
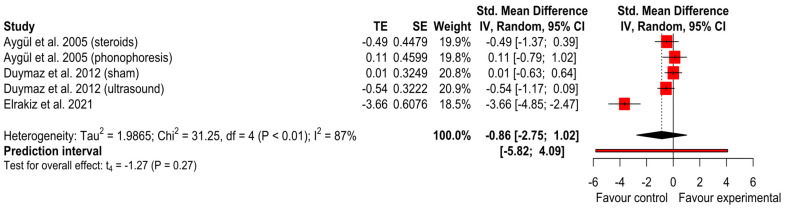
Forest plot of median sensory nerve conduction velocity [21,23,27].

**Figure 3 ijerph-20-04287-f003:**
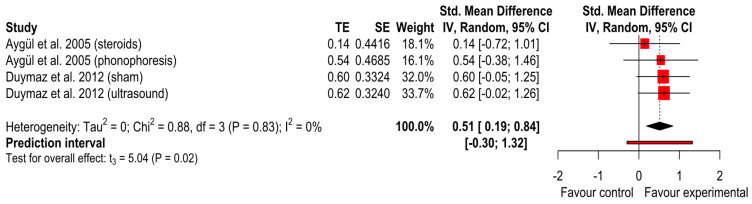
Forest plot of median sensory nerve amplitude [23,27].

**Figure 4 ijerph-20-04287-f004:**
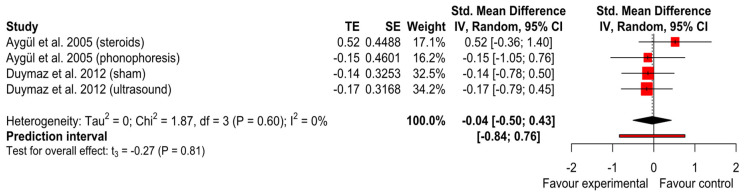
Forest plot of median sensory nerve distal latency [23,27].

**Figure 5 ijerph-20-04287-f005:**
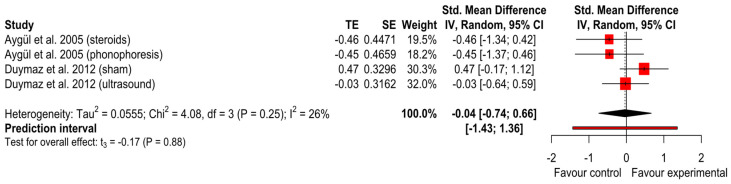
Forest plot of median motor nerve conduction velocity [23,27].

**Figure 6 ijerph-20-04287-f006:**
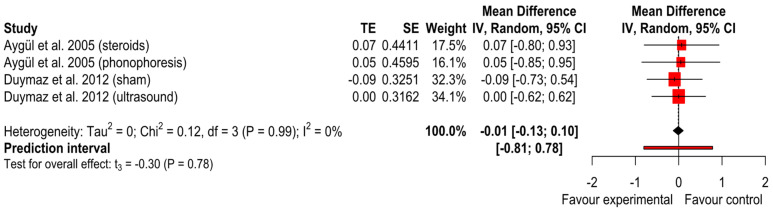
Forest plot of median motor nerve latency [23,27].

**Figure 7 ijerph-20-04287-f007:**
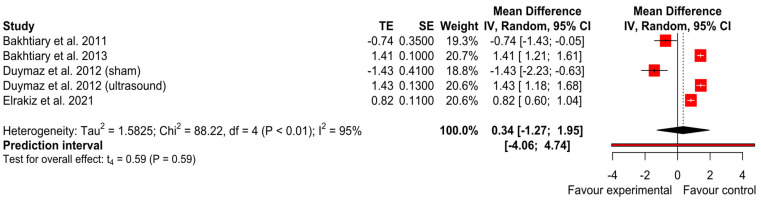
Forest plot of pain intensity (visual analogue scale) [21,22,23,25].

**Figure 8 ijerph-20-04287-f008:**
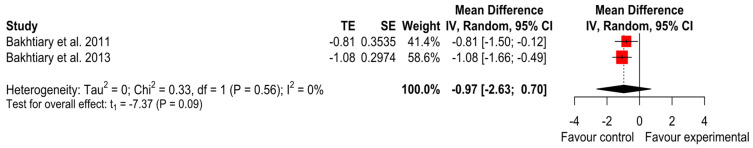
Forest plot of Handgrip strength [22,25].

**Figure 9 ijerph-20-04287-f009:**
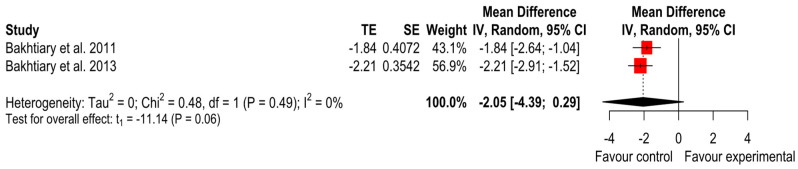
Forest plot of pinch strength [22,25].

**Table 1 ijerph-20-04287-t001:** Search strategy used in the different databases.

Search Strategy	Databases	Number of Records
Iontophoresis AND “carpal tunnel syndrome” Iontophoresis AND “carpal tunnel syndrome” AND treatment(s) Iontophoresis AND “median neuropathy” Iontophoresis AND “median neuropathy” AND treatment(s)	PubMed	78
	WoS	80
	Scopus	91
	CINHAL	39
	PEDro	8
“Iontophoresis” “carpal tunnel syndrome” “Iontophoresis” “carpal tunnel syndrome” “treatment(s)” “Iontophoresis” “median neuropathy” “Iontophoresis” “median neuropathy” “treatment(s)”	SciELO	0
	Total	296

**Table 2 ijerph-20-04287-t002:** Main characteristics of the included randomized clinical trials.

Authors (Year)/Study Design	Sample Size (Mean Age/SD)	Outcomes (Tools)	Interventions	Electrotherapy Parameters	Main Findings	PEDro Score (Appendix A)
Elrazik RKA et al., 2021 [21] RCT	N = 30 (21 women) IPH: n = 15 (30.6 ± 4.37) Shock waves: n = 15 (30.06 ± 2.86)	Electrophysiological assessment Pain intensity (VAS)	IPH: dexametasone + lidocaine Shock waves	IPH: Galvanic current. Drug administration on the active electrode placed over the CTS. 20 min/session, 3 sessions/week, 6 weeks Shock waves: 2000 pulses/session, 1.6 bars/session. 5 min/session, 3 sessions/week, 6 weeks	Both treatments showed significant changes after intervention. Significant differences favoring IPH compared to shock waves for pain and sensory velocity conduction were registered (*p* < 0.05)	7/10
Bakhtiary AH et al., 2013 [22] RCT	N = 34 (52 hands. 18 bilateral. 16 right-handed) IPH: n = 26 (48.2 ± 14.5) PHP: n = 26 (44.6 ± 12.8) CTS: mild to moderate severity	Electrophysiological assessment Pain intensity (VAS) Handgrip and pitch strength (Dinamometry)	IPH and PHP: dexamethasone sodium phosphate (0.4%)	IPH: Galvanic current. 2 mA/minute. Dose: 40 mA. 20 min PHP: 10 sessions (5 sessions/week). Pulse moder (25%). Frequency: 1 MHz. Intensity; 1 W/cm^2^. 5 min/session	PHP showed better results for all outcomes of study compared to IPH (*p* < 0.05)	9/10
Duymaz T et al., 2012 [23] RCT	N = 58 (55 women) IPH: n = 20 (51.85 ± 7.29) IPH + medication: n = 20 Ultrasound: n = 20 Sham IPH: n = 18 CTS: mild severity	Electrophysiological assessment Pain intensity (VAS) Range of movement during flexion–extension. Handgrip and pitch strength (Dinamometry) Semmes–Weinstein test Phalen and Tinnel test BQ and HAQ	Wrist Split and exercises (all groups) IPH: dexamethasone sodium diphosphate (0.4%) Ultrasound through subaquatic treatment Sham IPH: water	IPH: Galvanic current 2 mA. 20 min/session. Active electrode over the carpal tunnel. Passive electrode on distal part of the forearm. Sham IPH: electrodes with water on the same location Ultrasound: Continuos mode. 0.8 W/cm^2^. 5 min/session 5 sessions/week, 3 weeks (all groups)	The IPH showed more improvements compared with the other interventions (*p* < 0.05)	8/10
Gurkay E et al., 2012 [24] RCT	N = 54 (45 right-handed. 7 left-handed) Wrist splint: n = 18 (43 ± 6.9) IPH: n = 16 (44.1 ± 9.5) PHP: n = 18 (44 ± 8.7)	BQ Handgrip strength (Dinamometry) Hand dexterity (Nine-Hole Peg test)	Wrist splint (all groups): Neutral position IPH + Betamethasone (0.1%) PHP + Betamethasone (0.1%)	IPH: 3 weeks. (3 sessions/week). Galvanic current 4 mA. 10 min/session PHP: 3 weeks. (3 sessions/week). Frequency: Continuous mode. 1 MHz. Intensity: 1 W/cm^2^. 10 min/session	The three methods were effective Statistical significance between PHP versus wrist splint for BQ. No significant changes between groups for other outcomes	9/10
Bakhtiary AH et al., 2011 [25] RCT	N = 35 (51 hands) IPH: n = 19 PHP: n = 16 CTS: mild to moderate severity	Electrophysiological assessment Pain intensity (VAS) Handgrip and pitch strength (Dinamometry)	IPH: dexamethasone sodium phosphate (0.4%) PHP: dexamethasone (0.4%)	IPH: 2 weeks (1 session/week). Continuous current 0.4 mA/cm^2^. 10 min/session PHP: 2 weeks (1 session per week). Pulse mode. Frequency: 1 MHz. Intensity: 1 W/cm^2^. 5 min/session	PHP showed better results than IPH for handgrip strength (*p* = 0.006), pain intensity (*p* = 0.001), and sensory (*p* = 0.001) and motor (*p* = 0.008) electrophysiology	6/10
Amirjani N et al., 2009 [26] RCT	N = 20 (19 women. All bilateral cases) IPH: n = 10 (54 ± 10) Sham IPH: n = 10 (54 ± 10) CTS: mild to moderate severity	Electrophysiological assessment Levine Self-Assessment Questionnaire Semmes–Weinstein test	IPH: dexamethasone sodium phosphate (0.4%) Sham IPH: water	IPH: Continuous current 2 mA/min (Total dose 80 mA/minute). Active electrode over the carpal tunnel, passive electrode on the forearm Sham IPH: same but with water 3 sessions/week, 2 weeks; In both groups	No significant differences between groups studied (*p* > 0.05)	9/10
Aygül R et al., 2005 [27] RCT	N = 31 (56 hands. 31 women. 27 bilateral cases) Injection: n = 12 (46 ± 13.5) IPH: n = 9 (46.1 ± 13.5) PHP: n = 10 (44.1 ± 5.7) CTS: mild to moderate severity	Electrophysiological assessment BQ HAQ	Injection: dexamethasone sodium phosphate IPH: dexamethasone sodium phosphate (0.4%) IPH: dexamethasone sodium phosphate (0.1%)	Ionto: 3 weeks (5 session/week). Galvanic current 1 a 4 mA. 10 min/session PHP: 3 weeks (5 session/week). Frequency: 3 MHz. Intensity: 1 W/cm^2^. REA: 5 cm^2^. 10 min/session	The participants who received the injection showed better electrophysiological results than the other two groups. No statistical difference between IPH and PHP (*p* > 0.05)	7/10

BQ: Boston Questionnaire; CTS: carpal tunnel syndrome; N: Total sample size; n: number of participants per group; IPH: Iontophoresis.; PHP: Phonophoresis; HAQ: Health Assessment Questionnaire; REA: real emission area; VAS: visual-analogue scale.

## Data Availability

The data presented in this study are available on request from the corresponding author.

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
