# Peer review of "Use of Iontophoresis with Corticosteroid in Carpal Tunnel Syndrome: Systematic Review and Meta-Analysis"

_ijerph, 2023, doi:10.3390/ijerph20054287_

Round 1

Reviewer 1 Report

The design of the study does not raise my doubts for methodological reasons. What raises my doubts is that the authors recently published a very similar paper that overlaps substantially:

Martin-Vega, F.J.; Vinolo-Gil, M.J.; Perez-Cabezas, V.; Rodríguez-Huguet, M.; Garcia-Munoz, C.; Gonzalez Medina, G. Use of Sonophoresis with Corticosteroids in Carpal Tunnel Syndrome: Systematic Review and Meta-Analysis. J. Pers. Med. 2022, 12, 1160. https://doi.org/10.3390/jpm12071160

Both papers comparing methods of non-surgical treatment of carpal tunnel syndrome. In the comparative analysis, iontophoresis is also included. Studies share study design, methodology, search strategy etc. Both are structured similarly, and lead to similar conclusions.

Author Response

Dear reviewer 1,

On behalf of all authors, we are really glad for your time to review this manuscript and your comments. Regarding your question we design this systematic review after performing our first review, because we detected a gap in the literature. However, the literature found is really limited so some of the included studies are shared with the first review, this is one of the reasons that justify to reach similar conclusions. However, the aim of study and the search strategies are completely different. Both reviews are structured similarly because we work under a same structure for all systematic reviews (it is our work plan), we apologize for the inconvenience.

After the comments of the other two reviewers, we have made changes on our manuscript, primarily on the introduction and the conclusion. Please, we would be really glad if you could consider our work to be published in the notable International Journal of Environmental Research and Public Health. Once again, thank you.

Best regards.

Reviewer 2 Report

Introduction

The introduction should address and describe the various types of treatment and diagnosis that will be focus in the papers (IPH, shock waves, injections, PHP, ….).

Materials and methods

Indicate which treatments (interventions) were used to compare with ionotophoresis with corticosteroid.

Indicate which main findings were used.

Search strategy line 60

in Search strategy Why you did not use the term corticosteroid?

Table 2 Main findings you wrote sheck wave instead of shock wave

Line 287 you wrote eta-analyses instead meta-analyses

 Table 2 page 6 On main findings column “The IPH showed more improvements compared with the other interventions (p < 0.05)” – in this study you have IPH water and IPH dexamethasone

 Line 62 “The number of studies that used ultrasound as treatment for CTS is limited, currently the literature is using ultrasound more as a diagnostic tool than a therapeutic  option.” - This sentence makes no sense. Ultrasound for treatment and diagnosis are not comparable. Diagnostic ultrasounds are done for imaging and not for treatment, the machines are different.

 Strengths and limitations.

A major limitation of these articles used in the meta-analysis is that none of them used control groups with no treatment, or placebo control groups. Therefore, it cannot be said that the treatments have a positive effect. (Note: this may be understandable due to the fact that these are clinical articles, but we have to be aware of the conclusions we can draw).

Author Response

Dear reviewer 2,

Thank you for your comment. All suggestions helped us to improve our manuscript. Below you could find the response to your suggestions.

Best regards.

Introduction

The introduction should address and describe the various types of treatment and diagnosis that will be focus in the papers (IPH, shock waves, injections, PHP, ….).

## Response to introduction comment:

Thank you. Proper changes have been made in order to include the request information.

 Materials and methods

Indicate which treatments (interventions) were used to compare with ionotophoresis with corticosteroid.

Indicate which main findings were used.

# Response to comment of Materials and Methods:

Thank you for you comment. This information was clarified in our manuscript within the PICO model.

Search strategy line 60

in Search strategy Why you did not use the term corticosteroid?

# Response to comment about search strategy.

In a first search we include the term corticosteroid but the results were not suitable. Thus, considering the scarce number of results we decided to perform a more open search strategy to avoid losing potential records. We apologize for the inconvenience and we really thank your comment.

 Table 2 Main findings you wrote sheck wave instead of shock wave

# Comment about Table 2: We are really sorry for the mistake; we have made proper changes. Thank you for your comment. We have performed a review of English.

Line 287 you wrote eta-analyses instead meta-analyses

# Comment about Line 287: Again, we are really sorry. Proper changes have been made.

 Table 2 page 6 On main findings column “The IPH showed more improvements compared with the other interventions (p < 0.05)” – in this study you have IPH water and IPH dexamethasone

# Comment of page 6: Thank you for your comment. It is a mistake and proper changes have been made.

 Line 62 “The number of studies that used ultrasound as treatment for CTS is limited, currently the literature is using ultrasound more as a diagnostic tool than a therapeutic  option.” - This sentence makes no sense. Ultrasound for treatment and diagnosis are not comparable. Diagnostic ultrasounds are done for imaging and not for treatment, the machines are different.

# Comment of Line 62: We agree with your comment, so we removed this sentence.

 Strengths and limitations.

A major limitation of these articles used in the meta-analysis is that none of them used control groups with no treatment, or placebo control groups. Therefore, it cannot be said that the treatments have a positive effect. (Note: this may be understandable due to the fact that these are clinical articles, but we have to be aware of the conclusions we can draw).

# Comment about strengths and limitations:

We really appreciated your comment, so this part of the text has been rewritten following your indications. Thanks.

Reviewer 3 Report

very good paper, only two questions:

1. it would better to use hedge´s g as effect size measurement instead cohen´s d

2. why the forest plot for pain and functional outcomes are different, they didn´t show  two arm values, please clarify

Author Response

Reviewer 3:

very good paper, only two questions:

  1. it would better to use hedge´s g as effect size measurement instead cohen´s d
  2. why the forest plot for pain and functional outcomes are different, they didn´t show two arm values, please clarify

# Response to reviewer 3:

Dear reviewer 3,

Thank you for your nice comments which would help you to improve our manuscript. Below you could find the response to your suggestions. All results and forest plots were changed considering this modification.

Best regards,

# Response to comment 1:

We agree with your suggestion, so we have changed the effect size from Cohen’s d to Hedge’s g.  Thanks

# Response to comment 2:

This is a software issue. R studio could not perform the Mean Difference meta-analysis adding the mean values, because to calculates the mean differences you have to pre-calculate the effect size and then use the metagen function. The rest of the forest plots showed the mean values because the software allows to include the post-intervention values alone in the forest plot calculating the effect size using the metacont function. In order to show homogeneous forest plots, we changed Figure 2-9. Thanks for your comment.

Round 2

Reviewer 1 Report

Authors have made improvements in manuscript. Paper can be considered to be published in the International Journal of Environmental Research and Public Health.

Reviewer 2 Report

the authors answered the questions and updated the document